# Microstructure and Flexural Properties of Z-Pinned Carbon Fiber-Reinforced Aluminum Matrix Composites

**DOI:** 10.3390/ma12010174

**Published:** 2019-01-07

**Authors:** Sian Wang, Yunhe Zhang, Pibo Sun, Yanhong Cui, Gaohui Wu

**Affiliations:** 1College of Mechanical and Electrical Engineering, Northeast Forestry University, Harbin 150040, China; daowsa@nefu.edu.cn (S.W.); cyh_1995@126.com (Y.C.); 2Information Technology Department, Qingdao Vocational and Technical College of Hotel Management, Qingdao 266100, China; sunpibo@126.com; 3School of Materials Science and Engineering, Harbin Institute of Technology, Harbin 150080, China; wugh@hit.edu.cn

**Keywords:** metal–matrix composites (MMCs), carbon fiber, mechanical properties, z-pin reinforcement, laminate

## Abstract

Z-pinning can significantly improve the interlaminar shear properties of carbon fiber-reinforced aluminum matrix composites (Cf/Al). However, the effect of the metal z-pin on the in-plane properties of Cf/Al is unclear. This study examines the effect of the z-pin on the flexural strength and failure mechanism of Cf/Al composites with different volume contents and diameters of the z-pins. The introduction of a z-pin leads to the formation of a brittle phase at the z-pin/matrix interface and microstructural damage such as aluminum-rich pockets and carbon fiber waviness, thereby resulting in a reduction of the flexural strength. The three-point flexural test results show that the adding of a metal z-pin results in reducing the Cf/Al composites’ flexural strength by 2–25%. SEM imaging of the fracture surfaces revealed that a higher degree of interfacial reaction led to more cracks on the surface of the z-pin. This crack-susceptible interface layer between the z-pin and the matrix is likely the primary cause of the reduction of the flexural strength.

## 1. Introduction

Carbon fiber-reinforced aluminum matrix composites (Cf/Al) have high specific stiffness, high specific strength, good electrical conductivity and good fatigue properties, and thus have great potential in the aerospace and automotive industries [1,2,3]. While the traditional laminated plate structures have excellent in-plane properties, the lack of reinforcement in the thickness direction can cause delamination failure under thermal stress during the material forming process and the cutting force during machining processing [4]. Z-pinning is an effective technique for improving delamination resistance by inserting rods with high strength and high modulus such as titanium, steel, or fibrous carbon composite in the thickness direction of a composite [5]. Z-pins can enhance the interlaminar strength and impact damage tolerance of composites as well as the ultimate failure load and fatigue life of composite joints based on crack bridging forces [6,7,8,9]. Zhang et al. reported that the interlaminar shear strength of the Cf/Al composites could be increased by as much as 230% using the stainless steel z-pin [9].

Z-pinning increases the interlaminar mechanical properties but reduces the in-plane mechanical properties of the laminated composites [10,11,12]. Hoffmann and Scharr reported that z-pinning reduced the tensile strength of carbon fiber/epoxy laminates by 24–47% and the fatigue strength by 6–11% [10]. Knopp et al. found that using z-pins with a density of 2% and a diameter of 0.28 mm can reduce the flexural strength of the laminate by 27–31% [11]. Li et al. tested the compressive properties of laminates at room temperature and in dry, hot, and humid environments, and found that the z-pin reduced the compressive modulus by 17.2% and the compressive strength by 13.9% [12]. The reduction in the in-plane properties is attributed to the damage of the microstructure caused by the z-pin, including the waviness of in-plane fiber around the z-pin, fiber breakage, resin-rich pockets, and swelling of the z-pinned laminates [13,14]. The damage of these microstructures increases with increasing z-pin diameter and content, and thus the in-plane performance decreases with increasing z-pin diameter and content [10,14].

While the effect of z-pins on the in-plane mechanical properties of polymer matrix composites has been recognized, less is known about those of metal matrix composites. The interfacial layer composed of FeAl_3_ is formed by the interdiffusion of Fe and Al at the interface during the fabrication of z-pinned Cf/Al composites [15,16]. This strong interface can effectively transfer the interlaminar load from the matrix to the z-pin, resulting in improved interlaminar mechanical properties, but the brittle FeAl_3_ may adversely affect the in-plane mechanical properties of the composite [17]. Therefore, the degree of the interfacial reaction may be a key factor influencing the flexural performance of z-pinned Cf/Al.

Cf/Al composites are often subjected to flexural stress in typical applications. Thus, it is important to understand how the flexural strength changes at different z-pin parameters. This understanding could allow for better prediction and help maximize the performance of z-pinned Cf/Al. This work is an extension of earlier works by Zhang et al. [7,9] on the interlaminar shear strength and failure mechanism of z-pinned Cf/Al composites. The purpose of this paper is to study the influence of metal z-pin parameters on the flexural strength of z-pinned Cf/Al materials. Since the interface reaction is important for the mechanical properties of metal–matrix composites, the influence of the interface reaction degree on the strength and the failure mechanism of z-pinned Cf/Al is also discussed.

## 2. Experimental

### 2.1. Preparation of Z-Pinned and Unpinned Cf/Al Composites

The metal z-pin-reinforced Cf/Al composites were prepared using the pressure infiltration method. The matrix, in-plane reinforcement, and interlaminar reinforcement were 5A06Al, M40, and AISI321, respectively. The basic properties of M40 carbon fibers and AISI321 are listed in Table 1. The chemical compositions of 5A06 Al alloy and AISI321 are listed in Table 2 and Table 3, respectively. The flow chart of the manufacturing process of investigated composites is shown in Figure 1. In order to fabricate the preforms of carbon fibers, the carbon fibers were first unidirectionally wound by a winding machine (3FW250 × 1500, Harbin Composite Equipment Company, Harbin, China) to the desired shape. Then, the AISI321 z-pin was inserted into preforms of carbon fibers. The preforms of carbon fibers with z-pins were preheated to 500 ± 10 °C. The 5A06Al alloy was melted at 780 ± 20 °C and then infiltrated into the preforms under a pressure of 0.5 MPa. The pressure was carried on for 2 h to obtain Cf/Al composites. An unpinned composite was manufactured as a control martial under the same procedure to determine the damage to flexural performance caused by the z-pin. The volume fraction of the carbon fiber was 55%, which was measured by Archimedes method. 

### 2.2. Characterization Technique

Several types of z-pinned Cf/Al composite samples and unpinned Cf/Al composite samples were made for a three-point flexural test. The composites were reinforced by AISI321 stainless steel z-pins with a diameter of 0.3 mm, volume contents of 0.25%, 0.5%, and 1.0% in order to investigate the influence of the pin content on the flexural properties. Moreover, the composites were reinforced by AISI321 stainless steel z-pins with a volume content of 1% and diameters of 0.3 mm, 0.6 mm, and 0.9 mm in order to investigate the influence of the pin content on the flexural properties. The composites were machined into three-point flexural samples with dimensions of 60 mm × 10 mm × 2 mm and the z-pins aligned in parallel rows along the samples. The scheme and photo of the composite structure with the arrangement of the z-pins are shown in Figure 2. The row spacing between the z-pins of several types of z-pinned Cf/Al composite samples is listed in Table 4. The z-pin spacing can be determined using
(1)$$S=\frac{D}{2}\sqrt{\frac{\pi }{\rho \sin A}}$$
where S, *D*, *ρ*, *A* are z-pin spacing, diameter, volume content and angle, respectively. In this work, all z-pins were inserted vertically, with a z-pin angle of 90°. It can be seen that when fixing the z-pin volume content and angle, the z-pin spacing increases with the diameter. Fixing the z-pin diameter and angle, the z-pin spacing increases with the volume content. The three-point flexural test was performed on an Instron-5569 electronic universal tensile test machine according to GB/T232-2010 with a beam speed of 0.5 mm/min. A schematic of the test set-up is shown in Figure 3. The diameters of the cylindrical supports and cylindrical head were 5 mm and 10 mm, respectively. The support span was 40 mm (the span-to-thickness ratio was kept 20:1). The flexural strengths of unpinned and z-pinned composites were calculated using
(2)$$\sigma =\frac{3PL}{2{\mathrm{bh}}^{2}}$$
where σ is the flexural strength, *P* is the maximum load during the test, *L* is the support span, and b and h are the width and thickness of the sample, respectively. The microstructure of the composites was observed by a ZEISS 40MAT optic microscopy (OM, Carl Zeiss Microscopy, Jena, Germany). The fracture morphologies of samples were examined using a S-4700 scanning electron microscope (SEM, Royal Dutch Philips Electronics Ltd., Amsterdam, Netherlands).

## 3. Results and Discussion

### 3.1. Microstructure

Figure 4 shows the microstructure of the z-pinned Cf/Al composites. A distinct interfacial reaction layer was formed between AISI 321 steel and Cf/Al, indicating that the two had strong interfacial bonds. Zhang et al. [9] determined that the interfacial product formed during the sample preparation of z-pinned Cf/Al composite is FeAl_3_. As the metal pin diameter increased, the thickness of the interfacial reaction layer decreased. This means that the interfacial reaction of Cf/Al composites with thin metal z-pins was stronger than that of the Cf/Al composites with thick metal z-pins. This is because the interface layer was formed by the diffusion of Al and Fe atoms at a high temperature. When the aluminum liquid with high temperature was in contact with the z-pin during the preparation process, the heat of the aluminum liquid was transferred to the metal z-pin and caused the metal pin to heat up. The amplitude of the z-pin temperature increase depends on the quantity of heat absorbed. Larger z-pins require more heat to achieve the same increase in temperature. In other words, the thick z-pin takes more time to complete the heat exchange. During the sample preparation, the time limited the contact between the aluminum liquid and the z-pin, so the atomic diffusion was insufficient, and the thickness of the interface layer was small. A thin z-pin reached higher temperatures quicker. When the higher temperature is reached quickly, there is a longer time for atomic diffusion, and consequently, a higher amount of interfacial reaction occurs.

A distinct spindle region was formed around the z-pin. This region forms because the sample preparation process deform the fibers around the metal z-pin. During the waviness of the fiber, voids are formed, and these voids are filled with liquid aluminum in subsequent steps to form aluminum-rich zones characterized by fiber waviness, and aluminum enrichment. It is clear from the observations that larger z-pin diameters have a greater degree of carbon fiber waviness, and a larger area of aluminum-rich zones surrounding the z-pin. The mechanical damage to the fiber most likely occurred during z-pin insertion, although this is difficult to observe through the SEM.

### 3.2. Flexural Strength

Figure 5 shows the stress-deflection curves of the unpinned Cf/Al composite and the z-pinned Cf/Al composites with a diameter of 0.3 mm and volume fractions of 0.25%, 0.5%, and 1%. It suggests that the addition of the z-pin did not change the fracture behavior but did significantly reduce the flexural strength. The effect of the z-pin on the flexural strength is shown in Figure 6. It was found that the flexural strength of the composite decreased by 2%, 3%, and 25% after the z-pin was added. As the volume fraction of the z-pin increased, the flexural strength of the composite decreased, which is consistent with experimental results for z-pin-reinforced polymer matrix composites [11,14]. This is due to increased microstructural damage resulting from the increased volume fraction of the z-pin. By increasing the z-pin content, the degree of buckling of the carbon fiber increased and the overall area of the aluminum-rich zones also increased, leading to the dilution of the volume fraction of in-plane fiber. In addition, increasing the number of metal z-pins may increase the fiber breakage, and result in the loss of strength, although this was difficult to determine quantitatively.

Many studies have reported that when matrix-enriched areas become connected and a channel is formed due to the small z-pin’s spacing, crack expansion is more likely to occur, and the in-plane properties are further reduced [5,8]. As the z-pin content increases to 1%, small spacing between z-pins may be another reason for sudden drop in the strength of z-pinned Cf/Al composites.

Figure 7 shows the stress-deflection curves of the unpinned Cf/Al composite and the z-pinned Cf/Al composites with a volume fraction of 1% and diameters of 0.3 mm, 0.6 mm, and 0.9 mm respectively. The influence of the z-pin on the flexural strength is summarized in Figure 8 according to the stress-deflection curves. It can be seen that the addition of z-pins of diameters 0.3 mm, 0.6 mm, and 0.9 mm results in a 25%, 18%, and 13% reduction in flexural strength of the composites, respectively. This indicates that the knockdown of flexural strength decreased with the increase of the z-pin diameter, which is inconsistent with the experimental results of z-pinned polymer matrix composites [14]. For z-pinned polymer matrix composites, the deflection of carbon fibers and the dilution of carbon fiber content due to matrix-rich zones are the main factor for a reduction of the in-plane mechanical properties [12,13]. Thus, larger diameter z-pins lead to the greater deflection of carbon fibers and a larger area of matrix-rich zones, causing the lower flexural strength of the z-pinned polymer matrix composites. For the z-pinned Cf/Al composite, increasing the diameter causes the problems described above as well as other changes. Thin z-pin-enhanced Cf/Al has a more severe interface reaction than thick z-pin enhanced Cf/Al. Thin z-pin enhanced Cf/Al has a thicker interface diffusion layer due to the presence of a larger number of brittle reactants of FeAl_3_, and this layer becomes harder, more brittle, and more susceptible to fracture [9,16,17]. Thus, the degree of interfacial reaction is the main factor controlling Cf/Al flexural strength, and its effect is greater than that of damage to the microstructure. The thin metal z-pin-reinforced material has lower strength since it has minor microstructural damage but a high reaction degree. In addition, the thick z-pins have stronger fracture resistance, which results in the higher interlaminar strength of Cf/Al composites. Thus, when Cf/Al is required to have high flexural performance and delamination resistance, it is recommended to use relatively large metal z-pins (approximately 1 mm) for enhancement.

### 3.3. Fracture Surface

Figure 9 presents the fracture photograph of the unpinned Cf/Al composite and z-pinned Cf/Al composites with a metal pin volume fraction of 1% and diameters of 0.3 mm, 0.6 mm, and 0.9 mm. These images show that there are a large number of fibers pulled out from the fracture surface of the z-pinned Cf/Al composite (Figure 9a,c,e). This fracture feature is the same as that of the unpinned Cf/Al composite (Figure 9f). This similarity of fracture features indicates that the addition of the metal pin does not change the fracture mode of Cf/Al composites and that carbon fiber is still the main load carrier. These results are consistent with the results of the stress-deflection curve discussed in Section 3.2.

At the fracture surface of the z-pinned Cf/Al composites with diameters of 0.3 mm and 0.6 mm, a continuous shell-like interface reaction layer FeAl_3_ between the metal pin and aluminum was found (Figure 9a,c). The cracking of the metal pin/aluminum interface was along the FeAl_3_ interfacial reaction layer. There was no significant reaction layer with FeAl_3_ for the z-pinned Cf/Al fracture with a diameter of 0.9 mm (Figure 9e). This shows that the 0.3 mm and 0.6 mm metal z-pin reinforced Cf/Al composites were fractured along the cross-section containing the metal pin, while the 0.9 mm metal z-pin-reinforced Cf/Al composite did not fracture along the cross-section containing the metal pin.

Further high-magnification observations on the shell-like FeAl_3_ reaction layers revealed that the thick shell-like FeAl_3_ reaction layer of the z-pinned Cf/Al composite with a diameter of 0.3 mm was covered with microcracks (Figure 9b), whereas, less microcracks were observed in the thin reaction layer of the z-pinned Cf/Al composite with a diameter of 0.6 mm (Figure 9d). This shows that larger thicknesses of the reaction layer led to more of the brittle phase of FeAl_3_ and also resulted in more interface cracking. In addition, the thickness of the reaction layer not only affected the content of the brittle phase but also affected the crack growth. When the interface layer is thin, the crack size is also small, and it is more difficult to induce fiber breakage. That is, cracks are more likely to initiate and propagate when the reaction is severe. It is worth noting that the effects of interfacial reactions are not all harmful. Neither too heavy nor too small interfacial reactions are perfect. Achieving an appropriate amount of interface reaction is conducive to maintaining a good interface bonding strength and an effective load transfer from the matrix to reinforcement, therefore improving the interlaminar mechanical properties of composites. Nevertheless, in this work the interfacial reaction is relatively serious. After melted aluminum alloy was infiltrated into the preforms, the degree of interfacial reaction was dependent on the temperature of the preform and the melted aluminum alloy, and the diffusion time apart from contact area between the z-pin and the aluminum during the preparation process. Thus it is possible to decrease the temperature of the preform and melted aluminum alloy to optimize the z-pin/matrix interface, thereby minimizing the knockdown of flexural strength.

Based on the above analysis, the failure process of z-pinned Cf/Al composites was inferred. For 0.3 mm and 0.6 mm reinforced Cf/Al composites with a thick FeAl_3_ reaction layer, under stress in the fiber direction, the fracture occurred in the brittle FeAl_3_ phase first, which led to cracking of the interface. However, as carbon fiber was the main load-bearing phase, the composite did not fail. With increased loading, the cracks produced at the interface gradually extended into the carbon fibers, leading to the final failure of the sample, whereas, for the 0.9 mm z-pin-reinforced Cf/Al with a thin FeAl_3_ reaction layer, no significant interface cracks occurred, or at least interface cracks did not propagate into the fibers, due to good interfacial bonding. With further loading, the buckling fibers in the vicinity of the metal pin broke and eventually led to the overall fracture of the composite.

## 4. Conclusions

In the current work, the effect of the metal z-pin on the flexural strength of Cf/Al was investigated. Three-point flexural tests showed that the flexural strength of Cf/Al composites was reduced by 2–25% due to the introduction of the z-pin. The fracture surfaces revealed that the fracture mode of Cf/Al composites was not changed by the z-pins, and that carbon fibers contributed to the flexural strength of Cf/Al composites. The reduced flexural strength was attributed to the microstructural damage caused by the z-pin, such as waviness of the in-plane fiber, fiber breakage, aluminum-rich regions and formation of the brittle phase caused by the interfacial reaction between the metal pin and the matrix. The flexural strength declined with the increasing z-pin volume content due to greater damage to the microstructure, which is consistent with the results of polymer–matrix composites. The thick metal z-pin-reinforced Cf/Al composite caused more damage to the microstructure, but also had a lower interface reaction degree and less brittle phase, and thereby showed higher flexural strength than the thin metal z-pin reinforced Cf/Al composite. The study also showed that the brittle phase caused by the interfacial reaction is the main factor for the decline in the flexural strength of z-pinned Cf/Al composites. Thus it is expected that adjusting the process parameters, for example by decreasing the temperature of the preform and melted aluminum alloy, could possibly reduce the degree of interface reaction and thereby maximize the flexural strength of z-pinned Cf/Al composites.

## Figures and Tables

**Figure 1 materials-12-00174-f001:**
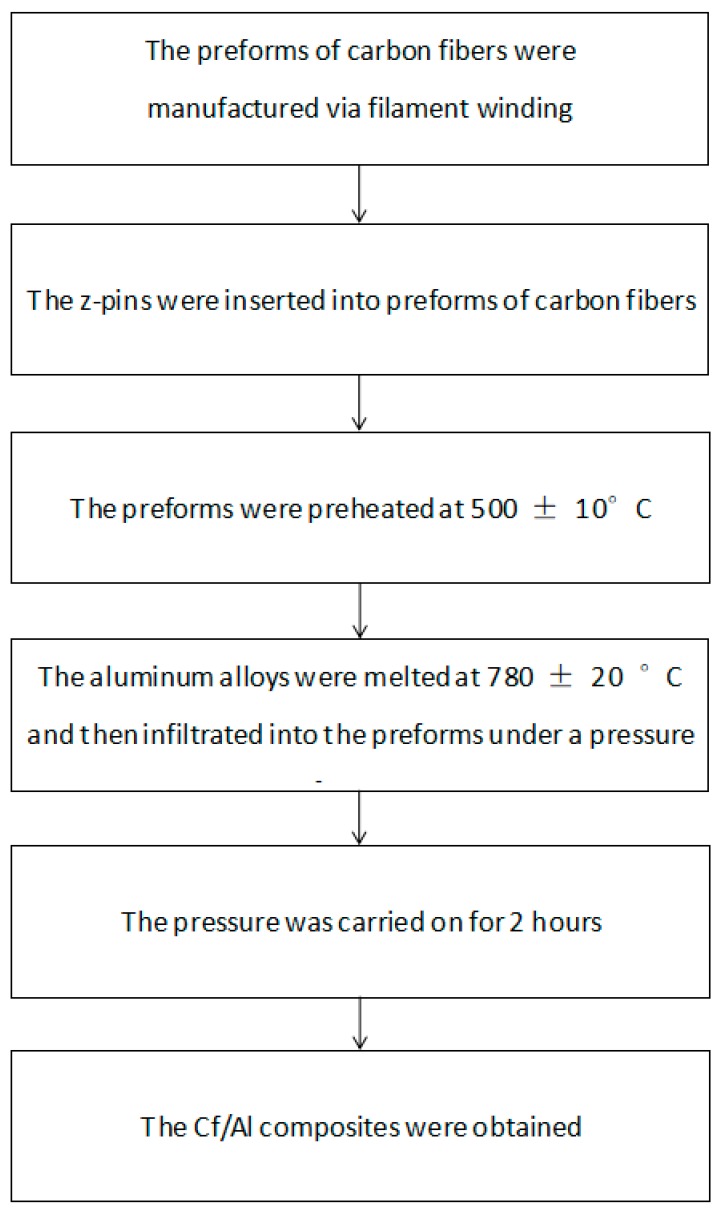
The flow chart of the manufacturing process of investigated composites.

**Figure 2 materials-12-00174-f002:**
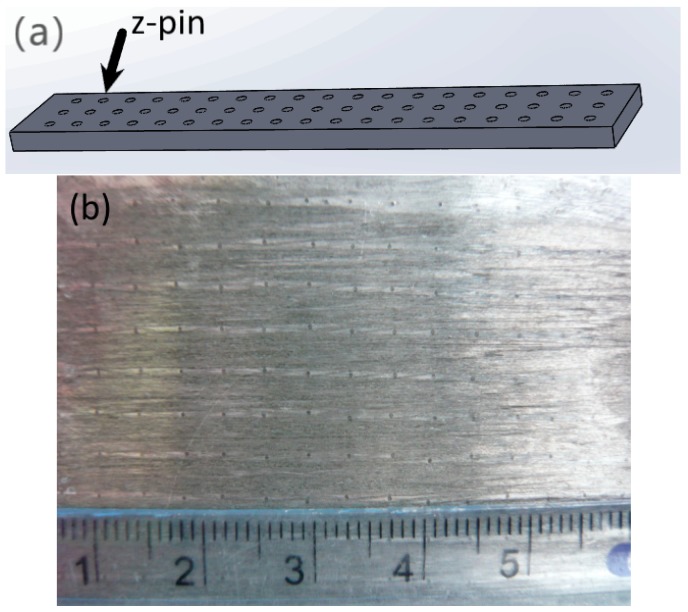
The (**a**) scheme and (**b**) photo of the z-pinned composite structure.

**Figure 3 materials-12-00174-f003:**
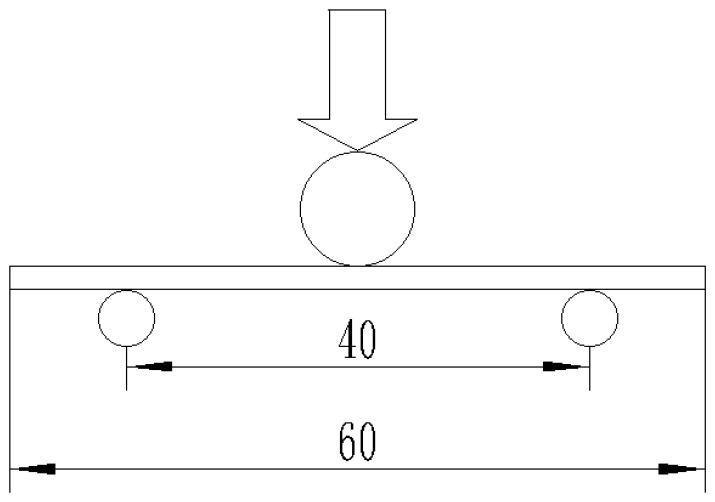
Scheme of the three-point flexural testing.

**Figure 4 materials-12-00174-f004:**
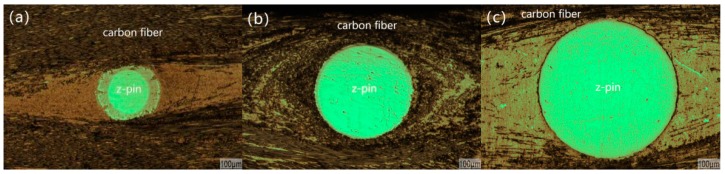
Microstructure of z-pinned Cf/Al composites. (**a**) ϕ0.3 mm metal pin, (**b**) ϕ0.6 mm metal pin, and (**c**) ϕ0.9 mm metal pin.

**Figure 5 materials-12-00174-f005:**
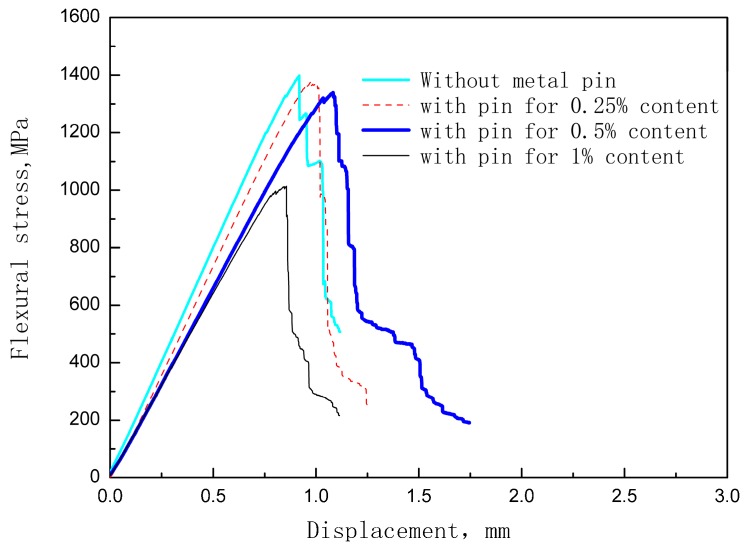
Typical Cf/Al stress-deflection results for samples without a z-pin and those with varying z-pin contents and a 0.3 mm diameter.

**Figure 6 materials-12-00174-f006:**
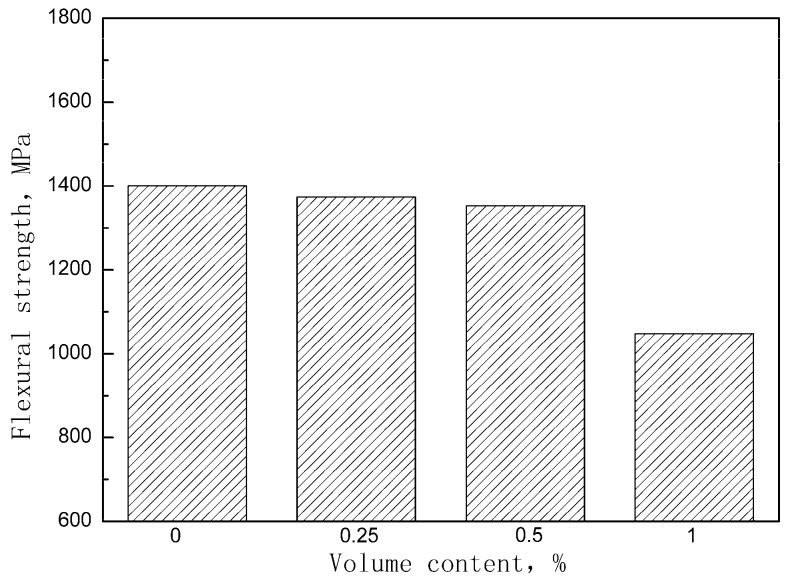
Effect of metal z-pin volume content on the flexural strength of the Cf/Al composite.

**Figure 7 materials-12-00174-f007:**
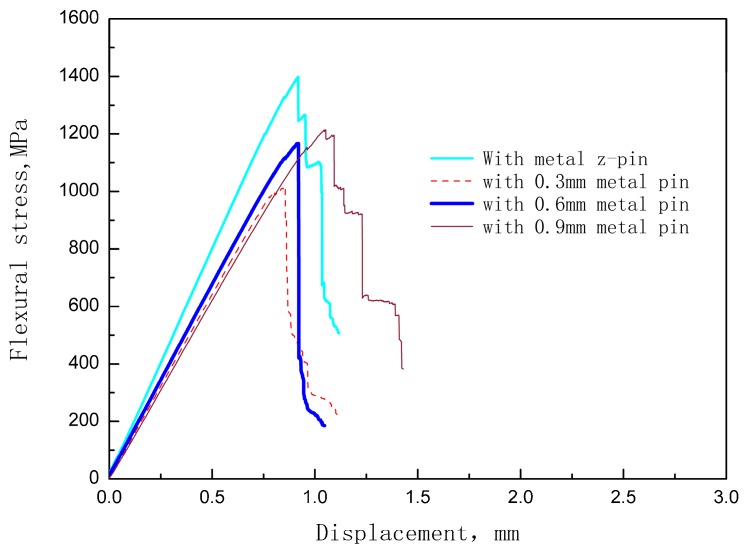
Typical Cf/Al stress-deflection results for samples without a z-pin and those with a z-pin content of 1% and different diameters.

**Figure 8 materials-12-00174-f008:**
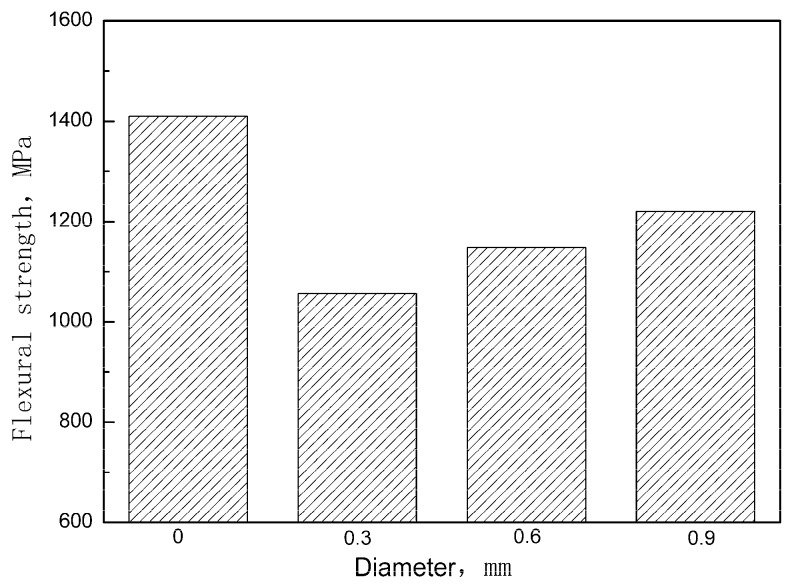
Effect of the diameter of the metal z-pins on the flexural strength of the Cf/Al composite.

**Figure 9 materials-12-00174-f009:**
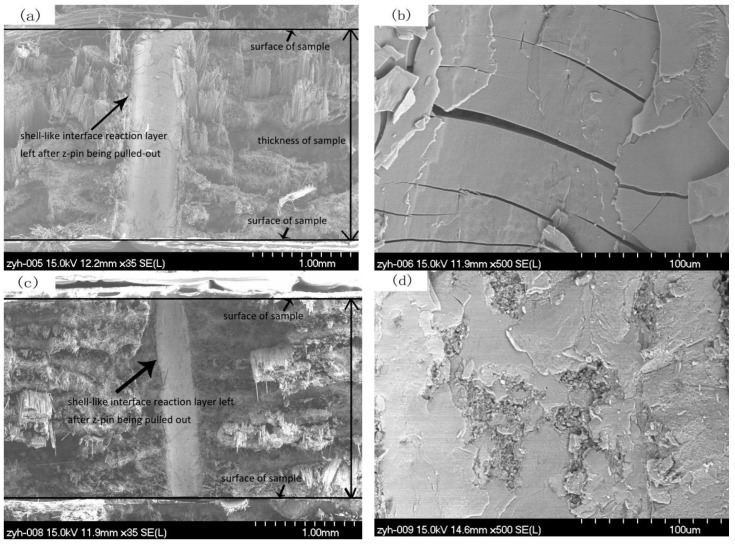
SEM images of the fracture surface of Cf/Al composites (**a**)–(**b**) with ϕ0.3 mm z-pin, (**c**)–(**d**) with ϕ0.6 mm z-pin, and (**e**) with ϕ0.9 mm z-pin, (**f**) without z-pin.

**Table 1 materials-12-00174-t001:** Basic properties of in-plane reinforcement and interlaminar reinforcement.

Materials	Tensile Strength (MPa)	Elastic Modulus (GPa)	Elongation to Fracture (%)	Density (g/cm³)
M40	4410	377	1.2	1.76
AISI321	1905	198	2	7.85

**Table 2 materials-12-00174-t002:** Chemical composition of 5A06 Al alloy (wt %).

Material	Mg	Mn	Si	Fe	Zn	Cu	Ti	Al
5A06 Al	5.8–6.8	0.5–0.8	0.4	0.4	0.2	0.1	0.02–0.1	Bal.

**Table 3 materials-12-00174-t003:** Chemical composition of AISI321 (wt %).

Material	Cr	Ni	Ti	Mn	Si	C	S	P	Fe
AISI321	17–19	8–11	0.5-0.8	<2.0	<1.0	<0.12	<0.03	<0.035	Bal.

**Table 4 materials-12-00174-t004:** The z-pin spacing of several types of z-pinned Cf/Al composite samples.

Z-Pin Volume Content (%)	Z-Pin Diameter (mm)	Z-Pin Angle (°)	Z-Pin Spacing (mm)
0.25	0.3	90	5.3
0.5	0.3	90	3.8
1	0.3	90	2.7
1	0.6	90	5.3
1	0.9	90	8.0

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
