# Peer review of "Microstructure and Flexural Properties of Z-Pinned Carbon Fiber-Reinforced Aluminum Matrix Composites"

_materials, 2019, doi:10.3390/ma12010174_

Round 1
Reviewer 1 Report
This is the work in which the effect of the z-pin on the flexural strength and failure mechanism of a Cf/Al composite with different volume contents and diameters of the z-pin was investigated. The article is a continuation of the authors group’ research previously described in [7] and [19] references.
The results are quite interesting for reader but the article contains some drawbacks that should be eliminated before its publishing.
Broad comments:
· In my opinion, the methodology part and presentation of the results should be improved.
· The “ microstructure” of the manufactured composite is poorly presented especially in comparison to authors previous papers, see [19] https://www.mdpi.com/1996-1944/11/10/1874/htm . In current form it is rather difficult to imagine the shape of the samples.
· The title of the article do not relates to microstructure hence it should be changed – or if authors should provide the overview of the composite (macroscopic photo, microscopic photo with visible reinforcement and z-pins location).
Specific comments:
· Authors wrote “important to understand how the flexural strength changes at different z-pin parameters This understanding could allow for better to predictions and optimize the performance of z-pinned Cf/Al.” – please give the criteria of optimization and write about the optimization conclusions.
· The experimental section is poorly written. Please explain the chemical composition of “grades” of used materials.
· Information given in L89-92 is not clear – I suggest to present the investigated composite structure in table form.
· What was the methodology used to estimation of % content of fibres (volume fraction)? – please explain it/describe it. Provide some structural photos.
· L75: explain what grade of “stainless steel” you meant and the same for aluminium alloy in L80.
· Please provide photos of the sample used in the test (flexural strength test).
· Please provide the scheme of composite structure, location of z-pins.
· Section 3.3 – that relates to fracture results – should contain the macroscopic photos of damaged samples and the overwide of the fracture.
· The figs. 2 -5 must be improved i.e. quality of the graphs is poor.
· L234. Author wrote “Therefore, controlling the degree of interface reaction by adjusting process parameters can optimize the interface to avoid in-plane performance reductions and provide better performance of Cf/Al composites” – in my opinion it is too general. Please explain it.
Author Response
Point 1: In my opinion, the methodology part and presentation of the results should be improved.
Response 1: Thanks for the reviewer’s comment. The author have revised the methodology part and presentation of the results.
...The metal z-pin reinforced Cf/Al composites were prepared by the pressure infiltration method. The matrix, in-plane reinforcement, and interlaminar reinforcement were 5A06Al, M40, and AISI321, respectively. The basic properties of M40 carbon fibers and AISI321 are listed in Table 1. The chemical compositions of 5A06 Al alloy and AISI321 are listed in Table 2 and Table 3, respectively. The flow chart of the manufacturing process of investigated composites is shown in Fig 1. In order to fabricate the preforms of carbon fibers, the carbon fibers were first unidirectionally wound by a CNC winding machine to the desired shape. Then, the z-pin was inserted into performs of carbon fibers. The preforms of carbon fibers with z-pins were preheated to 500 ± 10°C. The 5A06Al alloy was melted at 780 ± 20 °C and then infiltrated into the performs under a pressure of 0.5 MPa. The pressure was carried on for 2 hours to obtain Cf/Al composites. A unpinned composite was manufactured as a control martial under the same procedure to determine the damage to flexural performance caused by z-pin. The volume fraction of the carbon fiber was 55%, which was measured by Archimedes method.
Several types of z-pinned Cf/Al composite samples and unpinned Cf/Al composite samples were made for a three-point flexural test. The composites were reinforced by AISI321 stainless steel z-pins with a diameter of 0.3 mm volume contents of 0.25%, 0.5%, and 1.0% in order to investigate the influence of the pin content on the flexural properties. Moreover, the composites were reinforced by AISI321 stainless steel z-pins with a volume content of 1% and diameters of 0.3 mm, 0.6 mm, and 0.9 mm in order to investigate the influence of the pin content on the flexural properties. The composites were machined into three-point flexural samples with dimensions of 60 mm x 10 mm x 2 mm and the z-pins aligned in parallel rows along the samples. The scheme of composite structure with the arrangement of the z-pins is shown in Fig 2. The z-pin space of several types of z-pinned Cf/Al composite samples is listed in Table 4. The z-pin space can be determined using
S= (1)
where S, D, , A are z-pin space, diameter, volume content and angle, respectively. In this work, all z-pins were inserted vertically, z-pin angle is 90. It can be seen that fixing the z-pin volume content and angle, z-pin space increases with the diameter. Fixing the z-pin diameter and angle, z-pin space increases with the volume content. The three-point flexural test was performed on an Instron-5569 electronic universal tensile test machine according to GB/T232-2010 with a beam speed of 0.5 mm/min. The schematic of test set-up is shown in Fig 3. The diameters of cylindrical supports and cylindrical head are 5 mm and 10 mm, respectively. The support span is 40mm (the span-to-thickness ratio was kept 20:1). The flexural strengths of unpinned and z-pinned composites are calculated using
(2)
where is the flexural strength, P is the maximum load during the test, L is support span, b and h are width and thickness of the sample, respectively. The microstructure of composites was observed by a ZEISS 40MAT optic microscopy(OM). The fracture morphologies of samples were examined by a S-4700 scanning electron microscope(SEM)...
...Fig. 5 shows the stress-deflection curves of the unpinned Cf/Al composite and the z-pinned Cf/Al composites with a diameter of 0.3 mm and volume fractions of 0.25%, 0.5%, and 1%. It suggests the addition of the z-pin did not change the fracture behavior but did significantly reduce the flexural strength. The effect of the z-pin on the flexural strength is shown in Fig. 6. It was found that the flexural strength of the composite decreased by 2%, 3%, and 25% after the z-pin was added. As the volume fraction of the z-pin increases, the flexural strength of the composite decreases, which consistent with experimental results of z-pin reinforced polymer matrix composites...
... Fig. 7 shows the stress-deflection curves of the unpinned Cf/Al composite and the z-pinned Cf/Al composites with a volume fraction of 1% and diameters of 0.3 mm, 0.6 mm, and 0.9 mm respectively. The influence of the z-pin on the flexural strength is summarized in Fig. 8 according to stress-deflection curves. It can be seen that the addition of z-pins of diameters 0.3 mm, 0.6 mm, and 0.9 mm results in a 25%, 18%, and 13% reduction in flexural strength of the composites, respectively. This indicates that knockdown of flexural strength decreases with the increase of the z-pin diameter, which is inconsistent with the experimental results of z-pinned polymer matrix composites...
Point 2: The “ microstructure” of the manufactured composite is poorly presented especially in comparison to authors previous papers, see [19] https://www.mdpi.com/1996-1944/11/10/1874/htm . In current form it is rather difficult to imagine the shape of the samples.
Response 2: Thanks for the reviewer’s comment. We have added the scheme and macroscopic photo of sample with the arrangement of the z-pins (Fig 2) and revised the first paragraph in section 2.2 of Experiment.
Fig. 2. The (a) scheme and (b) photo of z-pinned composite structure
Point 3: The title of the article do not relates to microstructure hence it should be changed – or if authors should provide the overview of the composite (macroscopic photo, microscopic photo with visible reinforcement and z-pins location).
Response 3: Thanks for the reviewer’s suggestion. We have added a macroscopic photo of z-pinned composite. Moreover, the photos of fracture surface are also macroscopic photos. We re-edited the photos of fracture surface, where surfaces of samples and shell-like interface reaction layer left after z-pin being pulled-out were marked to make it clear.
Fig. 2. The (a) scheme and (b) photo of z-pinned composite structure
Point 4: Authors wrote “important to understand how the flexural strength changes at different z-pin parameters This understanding could allow for better to predictions and optimize the performance of z-pinned Cf/Al.” – please give the criteria of optimization and write about the optimization conclusions.
Response 4: Thanks for the reviewer’s suggestion. The authors have revised the third paragraph in section 3.3 of Results and Discussion according to your comments.
...That is, cracks are more likely to initiate and propagate when the reaction is severe. It is worth noting that the effects of interfacial reactions are not all harmful. Neither too heavy nor too small interfacial reaction is perfect. Achieving an appropriate amount of the interface reaction is conducive to maintain a good interface bonding strength and effective transfer of load from the matrix to reinforcement, therefore improves the interlaminar mechanical properties of composites. Nevertheless, in this work the interfacial reaction is relatively serious. After melted aluminum alloy was infiltrated into the performs, degree of interfacial reaction was dependent on temperature of preform and melted aluminum alloy and diffusion time apart from contact area between z-pin and aluminum during the preparation process. Thus it is possible to decrease the temperature of preform and melted aluminum alloy to optimize the z-pin/matrix interface, thereby to minimize the knockdown of flexural strength...
Point 5: The experimental section is poorly written. Please explain the chemical composition of “grades” of used materials.
Response 5: Thanks for the reviewer’s suggestion. We have added two tables of the chemical composition of “grades” of used materials (Tables 2 and 3).
Table 2. Chemical composition of 5A06 Al alloy (wt%).
Table 3. Chemical composition of AISI321 (wt%).
Point 6: Information given in L89-92 is not clear – I suggest to present the investigated composite structure in table form.
Response 6: Thanks for the reviewer’s suggestion. We have added a table of information of investigated composite structure (Table 4) .
Table 4. The z-pin space of several types of z-pinned Cf/Al composite samples
Point 7: What was the methodology used to estimation of % content of fibres (volume fraction)? – please explain it/describe it. Provide some structural photos.
Response 7: Thank you for your suggestion. The volume fraction of carbon fibers was measured by Archimedes method.
...The volume fraction of the carbon fiber was 55%, which was measured by Archimedes method...
Point 8: L75: explain what grade of “stainless steel” you meant and the same for aluminium alloy in L80.
Response 8: Thanks for the reviewer’s suggestion. The authors have revised the sentence and make it clear.
Point 9: Please provide photos of the sample used in the test (flexural strength test).
Response 9: Thanks for the reviewer’s suggestion. The authors have added a scheme of test set-up (Fig 3) and revised the second paragraph in section 2.2 of Experiment.
...The schematic of test set-up is shown in Fig 3. The diameters of cylindrical supports and cylindrical head are 5 mm and 10 mm, respectively. The support span is 40mm (the span-to-thickness ratio was kept 20:1). The flexural strengths of unpinned and z-pinned composites are calculated using
(2)
where is the flexural strength, P is the maximum load during the test, L is support span, b and h are width and thickness of the sample, respectively...
Fig. 3. The scheme of three-point flexural testing
Point 10: Please provide the scheme of composite structure, location of z-pins..
Response 10: Thanks for the reviewer’s suggestion. We have added a scheme of composite structure with the arrangement of the z-pins (Fig 2).
Fig. 2. The scheme of z-pinned composite structure
Point 11: Section 3.3 – that relates to fracture results – should contain the macroscopic photos of damaged samples and the overwide of the fracture.
Response 11: Thanks for the reviewer’s suggestion. The failure by overall flexural fracture of samples and other fracture features have been shown in photos of fracture surface. To make it clearer, we re-edited the photos of fracture surface.
Point 12: The figs. 2 -5 must be improved i.e. quality of the graphs is poor.
Response 12: Thanks for the reviewer’s suggestion. The authors have provided the new figures with high quality.
Point 13: L234. Author wrote “Therefore, controlling the degree of interface reaction by adjusting process parameters can optimize the interface to avoid in-plane performance reductions and provide better performance of Cf/Al composites” – in my opinion it is too general. Please explain it.
Response 13: Thanks for the reviewer’s suggestion. The authors have revised the sentences according to your comments.
...The thick metal z-pin reinforced Cf/Al composite with more damage to the microstructure has lower interface reaction degree and less brittle phase, thereby shows higher flexural strength. It shows as well, that brittle phase caused by interfacial reaction is the main factor for the decline in the flexural strength of z-pinned Cf/Al composites. Thus it is expected that adjusting process parameters such as decreasing temperature of preform and melted aluminum alloy could possibly reduce the degree of interface reaction thereby maximize the flexural strength of z-pinned Cf/Al composites...
Reviewer 2 Report
I consider the article interesting, especially as the subject related to the production of Al / Cf composites is still valid. However, because the title of the article indicates that it concerns the description of the structure, I suggest expanding this topic. Especially presenting the macrostructure and microstructure of the composite at lower magnifications. This will allow a more complete presentation of the structural characteristics of the composite material. I have noticed in the text the description indicating the volume fraction of carbon fibers. I also suggest expanding the description of the composite manufacturing process, it is not clear what was the factor causing the pressure of 90 MPa during the infiltration of the reinforcement structure.
I suggest make the corrections of the axle signatures in Figures 2-5 and the description of the legend in Figures 2 and 4. I have selected a few editing notes in the attached file

Author Response
Point 1: However, because the title of the article indicates that it concerns the description of the structure, I suggest expanding this topic. Especially presenting the macrostructure and microstructure of the composite at lower magnifications. This will allow a more complete presentation of the structural characteristics of the composite material.
Response 1: Thanks for the reviewer’s suggestion. We have added a macroscopic photo of z-pinned composite structure. Moreover, the photos of fracture surface are also macroscopic photos. According to your comments, we re-edited the photos of fracture surface, where surfaces of samples and shell-like interface reaction layer left after z-pin being pulled-out were marked to make it clear.
Fig. 2. The (a) scheme and (b) photo of z-pinned composite structure
Point 2: I have noticed in the text the description indicating the volume fraction of carbon fibers. I also suggest expanding the description of the composite manufacturing process, it is not clear what was the factor causing the pressure of 90 MPa during the infiltration of the reinforcement structure.
Response 2: Thanks for the reviewer’s comment. We have revised the section 2.1 of Experiment and make manufacturing process clear. Moreover, previously, the pressure during the infiltration was written as 90 MPa due to typo. In fact, the value of the pressure during the infiltration is 0.5 MPa.
... In order to fabricate the preforms of carbon fibers, the carbon fibers were first unidirectionally wound by a CNC winding machine to the desired shape. Then, the AISI321 z-pin was inserted into performs of carbon fibers. The preforms of carbon fibers with z-pins were preheated to 500 ± 10°C. The 5A06Al alloy was melted at 780 ± 20 °C and then infiltrated into the performs under a pressure of 0.5 MPa. The pressure was carried on for 2 hours to obtain Cf/Al composites...
Point 3: I suggest make the corrections of the axle signatures in Figures 2-5 and the description of the legend in Figures 2 and 4.
Response 3: Thanks for the reviewer’s suggestion. The authors have provided the new figures with high quality.
Point 4: I have selected a few editing notes in the attached file.
Response 4: Thanks for the reviewer’s suggestion. The authors have revised the sentences and list of references according to your comments.
...The 5A06Al alloy was melted at 780 ± 20 °C and then infiltrated into the performs under a pressure of 0.5 MPa...
Reviewer 3 Report
The manuscript entitled "Microstructure and Flexural Properties of Z-pinned Carbon fiber reinforced aluminum matrix composites reports out the findings of z-pinned carbon fiber reinforced aluminum matrix composites under flexural loadings. The content of the paper does not adds much to the previously known data, thereby the novelty of the paper remains average.
The article should be severely modified to be considered for acceptance in the materials journal like the experimental findings could be correlated with numerical or analytical methods.
Apart from that the figures of the current version is not clearly visible. thus the figure should be modified so that readers could be able understand the content of the paper more clearly. The visibility of the figures in the paper is very poor.
1> In experimental section, certain schematics for the sample preparation can be used to describe the creation of the specimens more clearly to the readers.
2>in Fig 6. It will be good to illustrate the location of the continuous shell-like interface reaction layer FeAl3 that the author is referring in the text on line 188 page 6.
3> In conclusions, the effect of the volume fraction of the Z pinned for a particular diameter pin on the flexural strength of composites can also be mentioned.
Author Response
Response to Reviewer 3 Comments
Point 1: In experimental section, certain schematics for the sample preparation can be used to describe the creation of the specimens more clearly to the readers.
Response 1: Thanks for the reviewer’s suggestion. The author have revised the manufacturing process part and added a flow chart of the manufacturing process of investigated composites (Fig 1) to make it clear.
...In order to fabricate the preforms of carbon fibers, the carbon fibers were first unidirectionally wound by a CNC winding machine to the desired shape. Then, the z-pin was inserted into performs of carbon fibers. The preforms of carbon fibers with z-pins were preheated to 500 ± 10°C. The 5A06Al alloy was melted at 780 ± 20 °C and then infiltrated into the performs under a pressure of 0.5 MPa. The pressure was carried on for 2 hours to obtain Cf/Al composites...
Fig. 1. The flow chart of the manufacturing process of investigated composites
Point 2: In Fig 6. It will be good to illustrate the location of the continuous shell-like interface reaction layer FeAl3 that the author is referring in the text on line 188 page 6.
Response 2: Thanks for the reviewer’s suggestion. We have re-edited the Figures and illustrated the location of the continuous shell-like interface reaction layer FeAl3.
Point 3: In conclusions, the effect of the volume fraction of the Z pinned for a particular diameter pin on the flexural strength of composites can also be mentioned.
Response 3: Thanks for the reviewer’s suggestion. We have revised the Conclusions according to your comments.
... The flexural strength declines with increasing z-pin volume content due to greater damage to the microstructure, which is consistent with the results of polymer matrix composites. ...
Reviewer 4 Report
The manuscript under review presents interesting results of the study of the carbon fiber reinforced composites. It was presented an original method of the specimens manufacturing with z-pins.
Authors described the influence of volume fractions and steel z-pins diameter on mechanical properties of the carbon fiber reinforced composites.
These results are of practical interest for the designers of new composite materials with carbon fiber due to the expansion of the knowledge base about the behavior features of the materials under load.
I would recommend adding an analysis of the relationship of the sizes and distribution of the pins in the composites volume.
As the remarks I would like to note that the authors presented facts that reduce the composites properties, but did not give recommendations for overcoming this disadvantage.
There are several grammatical errors and typos that authors can correct when re-checking.
Author Response
Response to Reviewer 4 Comments
Point 1: I would recommend adding an analysis of the relationship of the sizes and distribution of the pins in the composites volume.
Response 1: Thanks for the reviewer’s suggestion. We have added the an analysis of the relationship of the sizes and distribution of the pins in the composites and revised the section 2.2 of Experiment.
...The z-pin spacing of several types of z-pinned Cf/Al composite samples is listed in Table 4. The z-pin spacing can be determined using
S= (1)
where S, D, , A are z-pin spacing, diameter, volume content and angle, respectively. In this work, all z-pins were inserted vertically, z-pin angle is 90. It can be seen that fixing the z-pin volume content and angle, z-pin spacing increases with the diameter. Fixing the z-pin diameter and angle, z-pin spacing increases with the volume content...
Point 2: As the remarks I would like to note that the authors presented facts that reduce the composites properties, but did not give recommendations for overcoming this disadvantage.
Response 2: Thanks for the reviewer’s suggestion. We have added the analysis of reducing knockdown in flexural strength of z-pinned Cf/Al composites and revised the third paragraph in section 3.3 of Results and Discussion according to your comments.
...That is, cracks are more likely to initiate and propagate when the reaction is severe. It is worth noting that the effects of interfacial reactions are not all harmful. Neither too heavy nor too small interfacial reaction is perfect. Achieving an appropriate amount of the interface reaction is conducive to maintain a good interface bonding strength and effective transfer of load from the matrix to reinforcement, therefore improves the interlaminar mechanical properties of composites. Nevertheless, in this work the interfacial reaction is relatively serious. After melted aluminum alloy was infiltrated into the performs, degree of interfacial reaction was dependent on temperature of preform and melted aluminum alloy and diffusion time apart from contact area between z-pin and aluminum during the preparation process. Thus it is possible to decrease the temperature of preform and melted aluminum alloy to optimize the z-pin/matrix interface, thereby to minimize the knockdown of flexural strength...
Point 3: There are several grammatical errors and typos that authors can correct when re-checking.
Response 3: Thanks for the reviewer’s suggestion. The several grammatical errors and typos were corrected according to your comments.
Round 2
Reviewer 1 Report
Thank you for response. I accept all the explanations however I have some comments to the content of improved article:
1. Fig1 contains formatting symbols. Please improve it.
2. L126: Improve subscript in “FeAl3”.
3. Please improve in fig9 the (a-f) photos order.
Author Response
Response to Reviewer 1 Comments (Round 2)
Point 1: Fig1 contains formatting symbols. Please improve it.
Response 1: Thanks for the reviewer’s suggestion. The formatting symbols in the Fig 1 has been removed.
Point 2: L126: Improve subscript in “FeAl3”.
Response 2: Thanks for the reviewer’s suggestion. We have revised the sentence according to your comments.
...Zhang et al. [9] determined the interfacial products formed during the sample preparation of z-pinned Cf/Al composite is FeAl3...
Point 3: Please improve in fig9 the (a-f) photos order
Response 3: Thanks for the reviewer’s suggestion. We has corrected the typesetting.